# Effects of a Thermal Inversion Experiment on STEM Students Learning and Application of Damped Harmonic Motion

**Omar Israel González-Peña** [1,*,†], **Gustavo Morán-Soto** [2,†], **Rodolfo Rodríguez-Masegosa** [1,†] and **Blas Manuel Rodríguez-Lara** [1,†]

1  Tecnologico de Monterrey, Escuela de Ingeniería y Ciencias, Avenida Eugenio Garza Sada 2501, Monterrey 64849, N.L., Mexico; rrodolfo@tec.mx (R.R.-M.); BMLARA@tec.mx (B.M.R.-L.)
2  Department of Basic Sciences, Instituto Tecnológico de Durango, Blvd. Felipe Pescador 1830, Nueva Vizcaya, Durango 34080, Dgo, Mexico; gmorans@clemson.edu
*  Correspondence: oig@tec.mx or ogonzalez.pena@gmail.com
†  These authors contributed equally to this work.

**Abstract:** There are diverse teaching methodologies to promote both collaborative and individual work in undergraduate physics courses. However, few educational studies seek to understand how students learn and apply new knowledge through open-ended activities that require mathematical modeling and experimentation focused on environmental problems. Here, we propose a novel home experiment to simulate the dynamics of a flue gas under temperature inversion and model it as damped harmonic motion. After designing and conducting the experiment, twenty six first year students enrolled in STEM majors answered six qualitative questions to inform us about their epistemological beliefs regarding their learning process. Their answers imply that this type of open-ended experiments may facilitate students' understanding of physical phenomena and point to the significance of physics instructors as promoters of epistemological development. In general, students described this activity as a positive experience that helped them connect an environmental phenomenon with a fundamental physics concept.

**Keywords:** temperature inversion; damped harmonic motion; stem education; teaching methodologies; sustainable sevelopment goals; sustainable education; epistemology; mathematical modeling; educational innovation; higher education



## 1. Introduction

Finding strategies to promote student engagement in introductory physics courses is a challenge of our times. For instance, something as simple as identifying the preferences of students for learning physics by demonstrative problem solving on a blackboard or supervised independent collaborative work may improve their learning process [1].

In a traditional learning environment, introductory physics curricula is usually designed with the laboratory at the core of the learning process. It becomes a place to learn theoretical concepts as well as to conduct experiments [2,3]. This methodology increases student engagement in physics courses and improves conceptual understanding through manipulation of instruments and materials [3,4] to generate and process experimental data [5]. A well designed experiment may help STEM students develop self-regulated learning strategies [4,6,7], giving them the opportunity to build their own conclusions and boost their knowledge about physical phenomena and its interpretation [8]. Self-regulation and motivation is usually driven by epistemic beliefs [9,10] that describe the way students think about the nature of knowledge and knowing [11].

On the other hand, it is of utmost importance to develop Modeling Instruction (MI) plans to involve students beyond the four walls of the laboratory as engagement and cooperation play an essential role in the learning process of physics concepts [12]. These plans must improve the academic success of our students by boosting their interest

and involvement; for example, implementing active learning [13–15] using passive [16] or interactive [17,18] content as well as social media platforms [19] to promote students interaction with their whole learning environment within the framework of distance learning methodologies.

We believe that, in general, a well-designed instruction strategy facilitates STEM students understanding of physical phenomena that, in time, motivates the design and development of open-ended challenges aiming to provide solutions to urgent global issues. Here, we focus on the issue of air quality control and pollution to elaborate on the concept of simple and damped harmonic motion applied to particulate dynamics in the atmosphere. This issue is specially relevant for us as five Mexican cities (Mexico City, Monterrey, Guadalajara, Toluca and Leon) rank among the 13 cities with worst air quality according to a recent report from the Organization for Economic Co-operation and Development (OECD) [20], and it is common to observe pollutants trapped by temperature inversion in our daily life. Studying simple and damped harmonic motion applied to the dynamics of a particulate in the atmosphere could provide fertile ground to boost students cognitive process and to create awareness within the framework of the 2030 Agenda of Sustainable Development Goals. In Section 2, we state our purpose in detail and follow it with a literature review on teaching the damped harmonic oscillator in Section 3. Then, we present the details of our theoretical model and experimental setup in Section 4. Sections 5 and 6 show the methods and results of our qualitative research in detail, in that order. They are followed by a brief discussion in Section 7. Finally, we close with our conclusion in Section 8.

## 2. Purpose

Students learning new topics usually undergo an epistemological process where they reason about specific information obtained from different sources, then they claim knowledge of these new topics [21]. Kitchener's work suggests that open-ended problems or activities are more likely to engage students on an epistemological process than solving problems in class [22]. Our research aims to help physics educators by analyzing the effect of constructing and experimenting with a simulator of thermal inversion on students understanding of the damped harmonic oscillator.

Here, we propose a methodology to construct an isobaric troposphere simulator using air confined by a glass vessel where it is possible to introduce a foreign gas and make it oscillate by controlling the temperatures at the bottom and top ends of the container to simulate temperature inversion. We presented this activity to STEM students enrolled in first year introductory physics courses and asked them to collect experimental data of the dynamics to compare it with a numerical simulation of the damped harmonic oscillator. A goal for our students was to find values for the various parameters of the system that provide a good fit between experiment and theory. This was a collaborative activity for teams of four students that submitted a single project report. Individuals underwent an argumentative test that served as evidence to evaluate and accredit the understanding and mastering of a technical competence. In addition to this academic evaluation process, a cohort of students answered a questionnaire looking for their point of view on the effect of this experiment in their understanding of the thermal inversion phenomenon and its relation to damped harmonic motion.

## 3. Literature Review

The harmonic oscillator is at the core of our modeling of real-world devices involving integrated circuits, fluid mechanics, optical systems, and quantum technologies among others. Thus, engaging STEM students in an cognitive process that allows them to claim knowledge of this concept and extend its use beyond particular examples becomes a fundamental objective of physics education.

Pendulum and spring-mass systems are the standard textbook example of harmonic motion in introductory physics lectures. A spring-mass experiment is simple enough to introduce the idea of damped oscillations by measuring the position of the mass [23–27]. Conducting experiments with pendulum may improve student satisfaction under self-evaluation of their learning experience and knowledge of the subject [28]. Of course, real world devices are not completely harmonic; both spring-mass systems [29] and pendulum [30] beyond the small displacement or oscillation angle limit, in that order, serve as examples of basic non-linear models that undergraduate students may build using simple materials.

In addition to physics concepts, the oscillation of complex systems, like membranes or strings, allows the introduction of differential equations [31] and Fourier methods to physics problems [32]. In this direction, coupling a pair of simple harmonic oscillators may ease introducing the idea of coupled differential equations to students with some experience in classical mechanics [33]. From the simple to the complex, the harmonic oscillator offers an opportunity to understand the significance of mathematical modeling and visualize the effect of variable manipulation to engage STEM students into learning by physical interpretation [34].

Furthermore, analogies to the concept of harmonic oscillators may help students understand the workings of real-world devices and phenomena. For example, the infrared spectrophotometer [35] may be modeled as a spring-mass oscillator and the membrane vibration happening inside a microphone [36] is an analogy to a driven harmonic oscillator [37,38]. For more advanced courses, the idea of a classical harmonic oscillator may be extended to the quantum realm, for example, using basic calculus and algebra [39] or studying fluorescence in diatomic sulfide [40]. The motion of an electron in the presence of a two-dimensional potential is another simple example of harmonic motion [41] and analogies using spring-systems with coupled masses may help to introduce the formation of quantum bands to STEM students [42].

On the education side, the idea of interactive conceptual instruction using collaborative problem solving [43], computational simulations [44] and virtual laboratories [45], or alternative learning methods [46] show improvement clarifying misconceptions related to harmonic motion. Furthermore, flipped learning using software simulation of electrical circuits shows improvement in the students knowledge of the damped harmonic oscillator [47].

Our experimental proposal focuses on the importance of exploring systems beyond the spring-mass and pendulum in order to boost the epistemic cognitive process. In this direction, electronics present an opportunity to introduce highly controllable damping and non-linearities to harmonic oscillators beyond mechanical systems [48,49] and, for advance courses, the ability to produce, for example, time-dependent control to introduce continuous symmetries and its invariants following Noether theorem [50]. We may look into space and introduce the damped harmonic oscillator using the dynamics of particular celestial bodies [51] or to more complex setups, for example, lasers and optical resonators [52], classical gases confined by harmonic potentials [53], or the oscillation of a superconductor ring levitated by a magnetic field [54]. In particular, we are interested in fluids as they are exceptionally helpful to elucidate the effect of non-constant friction forces [55,56].

In the following, we present a simple experiment that allows STEM students to simulate an isobaric troposphere to explore, for example, the relation between damped harmonic motion and the dynamics of a particulate under temperature inversion at home.

## 4. Experimental Methods

The core of our proposal is a toy model of the troposphere to visualize how a gas cloud under temperature inversion behaves like a damped harmonic oscillator. Our experimental set-up, Figure 1, builds upon the idea that a transparent container whose ends are covered by a good thermal conductor may serve as a simulation of an isobaric troposphere at atmospheric pressure $p$ with control of bottom and top temperatures, $T_b$ and $T_t$. An inlet in

the bottom, a flue or exhaust, allows us to introduce a foreign gas, in our case the result of paper combustion, that we model as a non-interacting sphere with constant density $\rho_g$ in order to follow its center of mass motion. This approximation allows us to account the dynamics for the collection of combustion gases and particulates moving along with them as a whole.

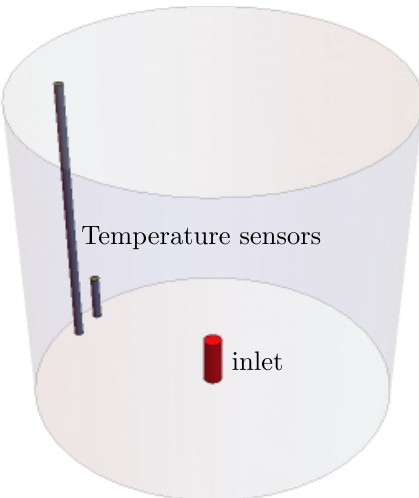

**Figure 1.** Sketch of the experimental setup used to simulate an isobaric troposphere. A clear container with two temperature sensors and an inlet to inject smoke.

Let us start with the model. Temperature control at the bottom and top ends of our simulation allows us to assume an air density that depends on the height. For the sake of simplicity, we suppose constant pressure and linear temperature gradient that allows us to approximate the air density,

$$\rho_{\text{air}}(y) = \rho_b(1 - c\,y), \qquad \text{with} \qquad c = \frac{1}{h}\left(1 - \frac{T_b}{T_t}\right), \tag{1}$$

up to first order on the height $y$, that we take as zero at the container bottom and $h$ at the top such that $y \in [0, h]$. We define the air density at the bottom and top ends,

$$\rho_b \equiv \rho_{\text{air}}(0) = \frac{pM}{RT_b} \qquad \text{and} \qquad \rho_t \equiv \rho_{\text{air}}(h) = \frac{pM}{RT_t}, \tag{2}$$

in that order, where our local atmospheric pressure at 540 m above the sea level is $p = 101{,}388$ Pa [57], the molar mass of air is $M = 28{,}965.4 \times 10^{-6}$ kg mol$^{-1}$ [58,59], the ideal gas constant is $R = 8.31447$ J mol$^{-1}$ K$^{-1}$, and the temperatures are given in Kelvin.

Assuming the collection of combustion gases and particulates moving along them as a non-interacting microscopic sphere allows us to model its center of mass dynamics using Newton second law,

$$m\frac{d^2y}{dt^2} = -w + F_B - b\frac{dy}{dt}, \tag{3}$$

where the forces in the right-hand-side of the equation are the weight $w = mg$ pointing downwards, the buoyant force $F_B = \rho_{\text{air}}V_{\text{gas}}g$ pointing upwards, and the Stokes drag for a sphere moving through a viscous fluid proportional to the drag coefficient, $b = 6\pi r_{\text{gas}}\eta_{\text{air}}$ in terms of the radius of the spherical particle $r_{\text{gas}}$ and the viscosity of the air $\eta_{\text{air}}$, and the velocity of the particle $dy/dt$.

The mass of the effective particle is given by its density and volume, $m = \rho_{\text{gas}} V_{\text{gas}}$, such that its weight becomes $w = \rho_{\text{gas}} V_{\text{gas}} g$ and the dynamics reduce to the following second order differential equation,

$$\frac{d^2 y}{dt^2} = \left[ \frac{\rho_{\text{air}}(y)}{\rho_{\text{gas}}} - 1 \right] g - \frac{b}{\rho_{\text{gas}} V_{\text{gas}}} \frac{dy}{dt}, \tag{4}$$

where we assumed that the gas density change induced by the temperature gradient is negligible at the time scale of the experiment and that the height of the container is small enough to produce no significant changes in the value of standard gravity. For the sake of simplicity, we assume that the viscosity of air has a negligible change with the temperature gradient. We use our linear approximation to the air density inside the container to unfold the model,

$$\frac{d^2 y}{dt^2} = -\frac{cg}{\rho_{\text{gas}}} y + \left( \frac{\rho_b}{\rho_{\text{gas}}} - 1 \right) g - \frac{9\eta_{\text{air}}}{2\rho_{\text{gas}} r_{\text{gas}}^2} \frac{dy}{dt}, \tag{5}$$

into that of a damped oscillator,

$$\frac{d^2 y}{dt^2} = -\omega_0^2 y + a_0 - \gamma \frac{dy}{dt}, \tag{6}$$

where the temperature difference controls the sign of the frequency,

$$\omega_0^2 = \frac{g}{h\rho_{\text{gas}}} \left( 1 - \frac{T_b}{T_t} \right). \tag{7}$$

Without considering the rest of the terms in the right-hand-side of the oscillator equation, if the temperature at the top is lower than that at the bottom, $T_t < T_b$, we have a positive squared frequency $\omega_0^2 > 0$ that yields a harmonic oscillator and we will see our gas sample oscillate. In the opposite case, $T_t > T_b$, we have a negative squared frequency $\omega_0^2 > 0$ that yields an inverted oscillator and our gas sample will rise and remain at the top of the container. For an isothermal simulation, $T_t = T_b$ the squared frequency is null, $\omega_0^2 = 0$ and only the external effective acceleration,

$$a_0 = \left( \frac{\rho_b}{\rho_{\text{gas}}} - 1 \right) g \tag{8}$$

has an effect on the dynamics. Without considering the rest of the terms in the right-hand-side of the equation, if the gas density is larger than that of the air at the bottom of our simulator, $\rho_{\text{gas}} > \rho_b$, the effective external acceleration is negative, $a_0 < 0$, and the gas sinks to the bottom of the container and stays there. If it is smaller, $\rho_{\text{gas}} < \rho_b$, the effective external acceleration is positive, $a_0 < 0$, and the gas rises to the top of the container and stays there. If they are equal, $\rho_{\text{gas}} = \rho_b$, the effective external acceleration is null and the gas does not move upwards nor downwards. Finally, the approximate drag frequency for the spherical particles of gas,

$$\gamma = \frac{9\eta_{\text{air}}}{2\rho_{\text{gas}} r_{\text{gas}}^2}, \tag{9}$$

where we assume a constant air viscosity as it changes from $\eta_{\text{air}}(T = 213.5 \text{ K}) = 0.0171 \times 10^{-3}$ Pa s to $\eta_{\text{air}}(T = 313.5 \text{ K}) = 0.0218 \times 10^{-3}$ Pa s for a temperature gradient of 100 K [60]. We take the average of these values as our constant viscosity, $\eta_{\text{air}} = 0.01945 \times 10^{-3}$ Pa s [60].

Our toy isobaric troposphere model allows the simulation of diverse dynamical phenomena. Temperature control at the ends of the simulator, for example, allows to switch the driven and damped oscillator between inverted or harmonic behaviour. Changing the atmospheric or foreign gases gives even more options to control and explore the parameters of the model. In the following, we focus our observations on temperature inversion.

Under standard conditions, the temperature gradually falls with the increase of altitude,

$$\Gamma = -\frac{dT}{dy}. \tag{10}$$

This is known as the thermal lapse rate; for example, the dry adiabatic lapse rate is around $\Gamma \approx 9.8 \times 10^{-3}$ K m$^{-1}$. Temperature inversion is the phenomenon that occurs when the thermal lapse rate $\Gamma$ changes sign from positive to negative; that is, a hot layer of air with low density hovers above a colder one with high density. In these situations, it is possible to observe smoke, or other pollutant gases, form a ceiling as the top low density layer of air stops their ascend.

In order to have a reference for the behavior, we present a numerical experiment in a container that is 0.15 m tall, take the standard value of gravity $g = 9.81$ m s$^{-2}$, the bottom of the container at room temperature $T_b = 298$ K leading to the density of air $\rho_b = 1.225$ kg m$^{-3}$ [61], the top of the container heated to $T_t = 373.15$ K and a constant viscosity of air $\eta_{\text{air}} = 1.945 \times 10^{-5}$ Pa s. We assign the gas a density of $\rho_g = 1.140$ kg m$^{-3}$ with a radius of $r_g = 6 \times 10^{-3}$ m that represents about 0.06% of the total volume of the container. These assumptions provide constants for the differential equation,

$$\omega_0 = 3.400 \text{ rad s}^{-1}, \tag{11}$$

$$a_0 = 0.731 \text{ m s}^{-2}, \tag{12}$$

$$\gamma = 1.567 \text{ rad s}^{-1}, \tag{13}$$

leading to the damped behaviour shown in Figure 2a. Figure 2b shows the effect of random variations on the temperatures, densities, effective particulate radius, air viscosity and initial velocity, $\{T_b, T_t, \rho_b, \rho_{\text{gas}}, r_{\text{gas}}, \eta_{\text{air}}, v_0\}$, following a normal distribution with mean value provided by the parameters above and standard deviation equal to one percent of the mean. We want to stress how such a small change in parameters produces a strong change in the dynamics.

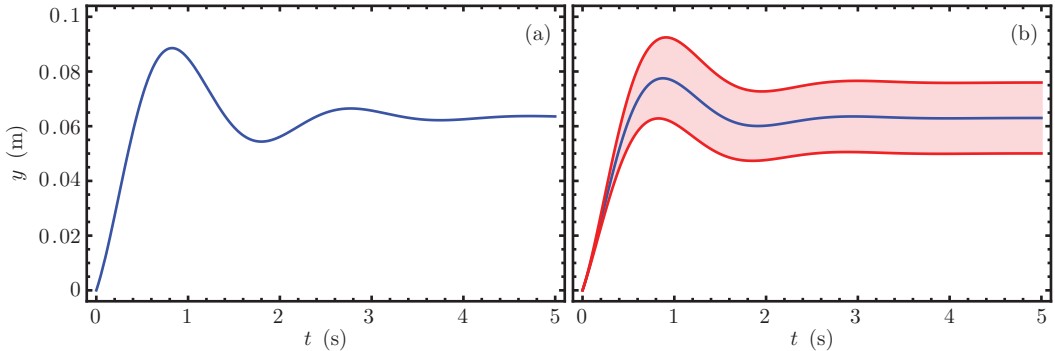

**Figure 2.** Damped oscillation for (**a**) a single particle of gas in the experiment with effective parameters provided by Equation (11) to Equation (13) and (**b**) mean value and region delimited by one standard deviation above and below the mean for a thousand random realizations with parameters following independent normal distributions.

We ask our students to reproduce the experimental setup, Figure 1, at home using a transparent tempered glass container to avoid fractures from temperature gradients; for instance, we use a coffee jar. In order to record the temperature at the bottom and top, we use two Vernier Stainless Steel Temperature Probes placed inside the glass container and a Vernier LabQuest Mini controller. These may be substituted by simple atmospheric

thermometers in contact with the external facet of the container at home. The sensors and a pewter straw to let smoke in are secured in place on the container lid using modeling clay to guarantee a good seal. We recommend securing a disposable plate to the container in order to hold ice cubes and secure access to the inlet straw that connects to a smoke container using plastic tubes; for instance, we used a candy jar to contain smoke from paper combustion but a party balloon or plastic bag may play the role. A light bulb lamp or an iron covered in aluminium foil may be used to change the temperature at the top of the container. Finally, we followed the effective center of mass dynamics of the flue gas using a logitech HD Pro Webcam C920 and Vernier Logger Pro but any given video capturing device and open source software like Tracker Video Analysis and Modeling Tool should do.

Figure 3 shows our experimental setup at home. As the container is not hermetically sealed, the pressure inside should be constant and equal to the atmospheric one. Our experiment allows the control of the input speed of the smoke as well as the bottom and top temperatures. We ask the students to experiment with these three parameters to explore the different dynamical regimes available. In particular, we ask for a detailed analysis of a case whose dynamics are an analogy to the damped harmonic oscillator. Their experimental data should allow them to fit for the dampening frequency $\gamma$ and the smoke density after figuring out the input velocity of the smoke without any temperature gradient. Figure 4 shows a sequence tracking of the approximated smoke cloud center of mass in an experiment with bottom and top temperatures in the order of 292.35 K and 306.35 K, respectively.

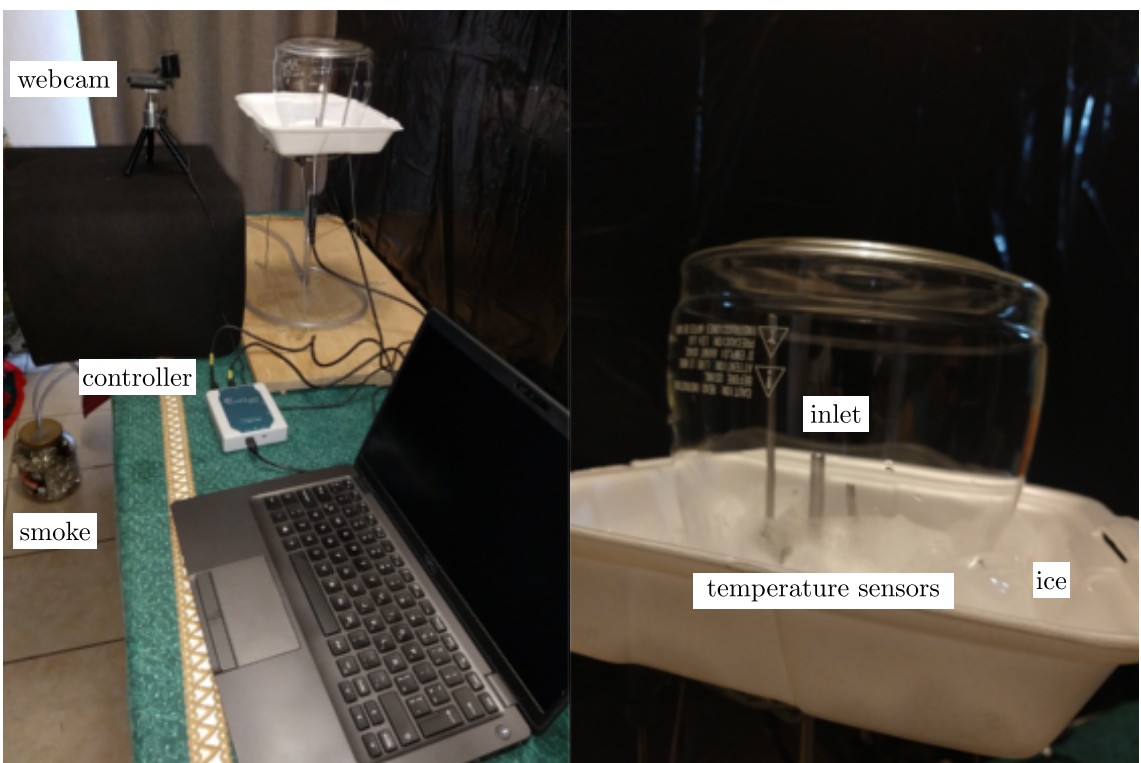

**Figure 3.** Experimental setup used to simulate temperature inversion in an isobaric troposphere.

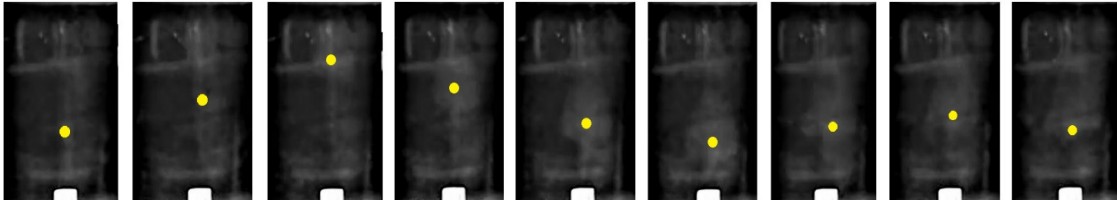

**Figure 4.** Example from an experimental measurement sequence. The yellow dot indicates the approximated center of mass of the smoke cloud.

Figure 5a shows experimental data points compared to the analytic model using an atmospheric pressure of $p = 102{,}300$ Pa and temperatures of $T_b = 292.75$ K and $T_t = 303.95$ K at the bottom and top of the container, respectively, leading to approximate air densities of $\rho_b = 1.218$ kg m$^{-3}$ and $\rho_t = 1.173$ kg m$^{-3}$ accounting to a height difference of $h = 0.08$ m between the temperature sensors. We assume the smoke density of about $\rho_{\text{gas}} = 1.210$ kg m$^{-3}$ with radius $r_{\text{gas}} = 5.97 \times 10^{-3}$ m and initial velocity of the order of $v_0 = 0.145$ m s$^{-1}$ leading to an effective acceleration, $a_0 = 0.066$ m s$^{-2}$, as well as effective oscillator and damping frequencies $\omega_0 = 1.93$ rad s$^{-1}$ and $\gamma = 2.027$ rad s$^{-1}$, in that order. The difference between experimental data and the dynamics under our educated guess may be due to variations in any of the assumptions, as we discussed before, and provides an opportunity for discussion. Our ansatz provides a good starting point for a better fitting using, for example, Newton least squares method or Levenberg-Marquardt method for nonlinear least squares yield an effective acceleration, $a_0 = 0.053$ m s$^{-2}$, effective oscillator and damping frequencies $\omega_0 = 2.025$ rad s$^{-1}$ and $\gamma = 2.137$ rad s$^{-1}$, in that order, and initial velocity $v_0 = 0.156$ m s$^{-1}$ that provides a better fit shown in Figure 5b.

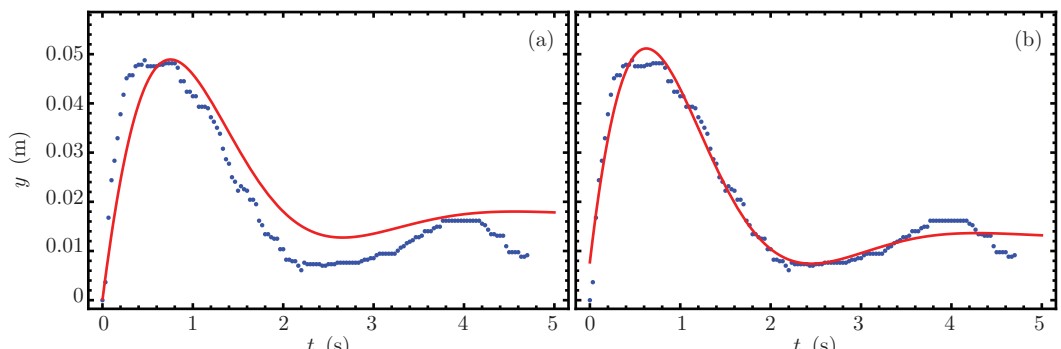

**Figure 5.** Example from an experimental measurement sequence (data points) and its corresponding fit (solid line) from (**a**) theoretical parameters and (**b**) a numerical fit using these parameters as starting point.

## 5. Qualitative Research Methods

In the following, we address the qualitative methods used to analyze our students point of view regarding the effect of this learning activity on their understanding of the damped harmonic oscillator.

### 5.1. Participants

We selected a cohort of 26 students from a total population of 185 individuals that conducted the experiment while enrolled in different groups of the same five-week long introductory physics module during the Fall 2020 semester. The modules were lead by instructors that voluntarily accepted to distribute our questionnaire with six open-ended questions at the end of the experiment. Students were offered extra credit if they decided to answer the questionnaire; all of these 26 students answered the six questions and we collected their answers before the module ended. Our students were in the first semester of diverse STEM majors at Tecnologico de Monterrey where they are required to partici-

pate in challenges related to real-world issues as part of their professional competencies development [3,62–64].

### 5.2. Data Collection

Our thermal inversion experiment aims for our students to explore damped harmonic motion in a unique way to try and help them understand this concept. We distributed a questionnaire after they finalized the experiment in order to analyze the effect of our open-ended activity on their understanding of the damped harmonic oscillator. The six questions from the questionnaire were adapted from the Engineering Related Beliefs Questionnaire (ERBQ) [65]. The ERBQ aims to measure the beliefs of students regarding the nature of STEM-related knowledge and knowing and it has proved a successful tool for the analysis of STEM students epistemic process after solving open-ended problems [66]. These six selected questions are related to open-ended problem solving aspects of the cognitive process of STEM students. They were translated to Spanish language and then adapted to our thermal inversion experiment context to ask students about the certainty and sources of their physics knowledge. A researcher with experience teaching and conducting education research in both Spanish and English at Mexican and U.S. universities made the translation. The experience of this researcher facilitated the interpretation and accurate translation of the meaning of every question. The aim was to adapt those questions in order to reflect the context of Mexican students in their physics courses and avoid possible cultural misunderstandings [67]. Additionally, four physics instructors provided feedback about the clarity and the content of the questions aiming to reconcile possible misinterpretations. Their comments helped to establish content and face validity for the questionnaire [68]. The questionnaire was back translated to English language by the same researcher for the purpose of this paper, see Appendix A.

### 5.3. Data Analysis

The four researchers involved in this study conducted the qualitative analysis. We analyzed the data collected with the questionnaire using open coding to let emerging codes to stay as close as possible to the own words and ideas of the students [69]. The final codes for each student were compared side by side with codes from other students aiming to find similarities that could be coded together into meaning units [70]. This coding philosophy is similar to the methodology proposed in the constructivist grounded theory [71]. This qualitative analysis approach helped us draw conclusions on how our thermal inversion experiment influenced our students knowledge and understanding of the damped harmonic oscillator. It also helped us ensure that these meaning units appropriately reflected the responses and feelings about the experiment of our students. We used the epistemical belief system framework proposed by Schommer-Aikins [72] as a lens to analyze and compare these meaning units. This framework helped us determine how the epistemological beliefs of our students evolved during the learning process and how they developed and acquired new knowledge. At the end, we selected some of the most relevant comments from the students to support our findings. These comments were translated from Spanish to English language by the same researcher that translated the questionnaire and are included in the following section.

## 6. Qualitative Results

Half of our students (13) reported using a single methodology or theory for designing and developing their thermal inversion experiment, while the other half (13) used a combination of two or more different methodologies or theories for their design. Regardless of whether they used only one methodology or different methodologies, our students reported that the most likely starting point for their thermal inversion experiment design was the methodology previously explained by their instructor. Student A4 noted: "I thought of some less orthodox methods but I mostly focused on the one taught in the lessons."

Most students (20) reported asking for help to their instructor during the design and development of their thermal inversion experiment. Seventeen students reported looking for more than one source of information; for example, asking other instructors, classmates, or family members to confirm and complement the information about the damped harmonic oscillation they had. Student A19 noted: "We asked the challenge's instructor for help. However, we needed further help and requested it from friends in a senior class." Only one student reported completing the inversion experiment without asking for help.

When students were asked if they searched for additional sources of information to complement what they knew about the damped harmonic oscillator, almost all of them (25) decided to look for additional information in different sources outside of what they learned in their course. Most of these students (17) looked for more than one source of external information. The most recurrent sources of external information were videos (15) and websites (13). Other less common sources were textbooks (5) and scientific papers (3). Eight students reported looking for only one external source of information. Only one student decided not to look for external sources of information.

Almost all students (24) reported that the thermal inversion experiment helped them to better understand the damped harmonic oscillator. These students stated that this experiment helped them clarify some doubts about damped harmonic motion, while they learned different ways to apply this topic in real life problems. Student A23 noted: "It was really helpful. Before, I thought that harmonic movement only applied to springs, but now I understand that it also applies to more complex systems such as fluids." On the other hand, only two students mentioned that this experiment was confusing and it did not help them understand the damped harmonic oscillator. Most of our students (23) stated that the thermal inversion experiment could have different final results that could be valid depending of the methodology used during the design, or some differences in the obtained measurements; seven of these students argued that their responses need to be supported by theory to be considered a valid response.

Half of our students (13) stated that the thermal inversion experiment might have better results in their understanding of the damped harmonic oscillator if the instructor could spend extra time explaining the theory and giving them more details about how to apply this concept to solve different problems in different contexts. This issue was more evident for seven students that mentioned having struggles to answer the argumentative test due to difficulties adapting their knowledge and experiences from our thermal inversion experiment to the context of the text. Student A11 noted: "I felt prepared to answer questions related to the challenge, but the argumentative exam had nothing to do with thermal inversion."

## 7. Discussion

Students in our research showed that they expected instructions from their professors in order to design the experiment following what they have previously learned and practiced in classes. They were open to follow different methodologies to design and develop their experiment, but they stated that the guidance from an expert is the key to success in this type of open-ended activities. This may be related to a low level of epistemological maturity for college level students [73], where they need to learn new knowledge and ways to apply it from an epistemological authority that is considered the only source of reliable information [11]. Most of our students asked for help during the experiment design and development, showing they are likely to search for an epistemological authority that could help them to develop their knowledge during the experiment process as well. Students that asked for help reported going directly to their instructors with specific doubts and questions. This behavior pinpoints the importance of physics instructors and the information they provide to their students before and during the development of the experiment.

Students reported that they are likely to seek additional information sources to analyze what they know and solve some doubts; videos and websites are the most common places where they look for more information. This interest in seeking additional sources of knowledge may be seen as a self-regulated learning action [10]. Promoting these types of actions may ultimately help students understand class material and perform better in solving open-ended problems. Although most students searched for additional sources of information, very few consulted scientific publications and textbooks to expand their knowledge and answer questions. This lack of interest in scientific publications is an area of opportunity to improve the research skills of our students and should be considered by physics instructors when advising their students on the advantages of seeking information supported by scientific evidence [74]. Physics instructors should be aware that their students are likely to search for the most accessible source of information, and then build their knowledge from that source rather than looking for more reliable sources which may be harder to find and analyze to construct their knowledge [75].

At the end of the thermal inversion experiment and the argumentative test, almost all students described this activity as a positive experience that helped them to better understand the damped harmonic oscillator and how this physical concept may be applied in different contexts. This type of open-ended activities may help students develop their knowledge and make them think about different sources where they can look for information to solve their doubts. Giving the opportunity to experience these type of experiments to students may help physics instructors facilitate the epistemological development of their students; different studies have shown similar results for different courses and contexts [66,76]. Reaching higher levels of epistemological maturity may benefit the development of students as critical thinkers, making them more likely to think that the development of their knowledge is their responsibility and that they need to search and confirm their own knowledge and beliefs [73]. This is relevant for physics instructors because some students stated that they felt the knowledge about thermal inversion that they learned and experimented was not transferable to different instances involving the damped harmonic oscillator. This lack of abilities to apply the same physics principle to solve different problems is common in low levels of epistemological maturity [73]. This position on how to learn and apply new concepts could hinder the possibilities of our students to understand complex physical phenomena [77].

## 8. Conclusions

It is important that physics instructors strive to provide enough information to their students so that they develop sufficient confidence to successfully complete open-ended activities like our thermal inversion experiment. The process of preparing students with the basic knowledge to perform experiments such as the one presented here needs certain precautions. Physics instructors must provide enough information without leading the entire experience so that their students have the opportunity to solve emerging issues on their own. They must take the role of facilitators, holding their students responsible for solving their own doubts by searching for reliable sources of information that help them develop their knowledge and solve their doubts. Our thermal inversion experiment may be used by instructors to reinforce students knowledge of the damped harmonic oscillator and to motivate them to evolve from low to mature levels of epistemological maturity where they are more likely to seek and develop their own knowledge. It may also help students develop research interest and skills leading to a deeper understanding of scientific concepts in advanced semesters. In addition, the possibility of controlling the various parameters in our isobaric troposphere simulator allows for the real-time visualization of different dynamics that may be useful in courses in Earth Science or Environmental Engineering. This type of experiment opens a window of opportunities for faculty to propose more challenging activities that might facilitate facing the great issues that society has related to the 2030 Agenda of Sustainable Development Goals.

**Author Contributions:** All authors contributed equally to this work. All authors have read and agreed to the published version of the manuscript.

**Funding:** This research was funded by Tecnologico de Monterrey grant number NOVUS-2020-308.

**Institutional Review Board Statement:** Ethical review and approval were waived for this study by Tecnologico de Monterrey.

**Informed Consent Statement:** Informed consent was obtained from all subjects involved in the study.

**Data Availability Statement:** Not applicable.

**Acknowledgments:** O.I.G.-P. acknowledges to Arath Marín-Ramírez for the literature collecting process on the previous studies in harmonic oscillator in the education context. R.R.-M. is grateful to his students for agreeing to share their evaluations anonymously for the academic analysis of this work. B.M.R.-L. acknowledges fruitful discussion and support from Benjamin Raziel Jaramillo Ávila. O.I.G.-P., R.R.-M. and B.M.R.-L. acknowledge NOVUS, the Writing Lab, Institute for the Future of Education, Tecnologico de Monterrey, Mexico, for the financial and logistical support.

**Conflicts of Interest:** The authors declare no conflict of interest. The funders had no role in the design of the study; in the collection, analyzes, or interpretation of data; in the writing of the manuscript, or in the decision to publish the results.

## Abbreviations

The following abbreviations are used in this manuscript:

| | |
|---|---|
| MDPI | Multidisciplinary Digital Publishing Institute |
| DOAJ | Directory of open access journals |
| TLA | Three letter acronym |
| LD | linear dichroism |

## Appendix A. Questionnaire

1. When preparing your answers on the thermal inversion experiment, did you consider several methods to answer the questions or did you just consider one method to find the solution? Please explain the reasoning that led to the methods that you used.

2. Did you ask for help to answer the questions? If so, who did you ask (for example, instructors, classmates, friends, family, or tutors)?

3. When answering the exercises or preparing the report, did you check any additional source of information (such as videos, books, or tutorials) besides the data given to you by the instructors? If so, which sources did you check?

4. Do you think that experimental activities based on the application of a physical concept (such as the activity on damped movement) helps improve your understanding on thermal inversion? Please explain the reasoning behind your answer.

5. Do you think that the questions you ask yourself during the experiment only have one correct answer, or that there could be more than one correct answer depending on your interpretation of certain variables and data? Please explain the reasoning behind your answer.

6. Did you feel that you had sufficient background knowledge to understand the thermal inversion experiment and to correctly answer the argumentative exam? If not, please explain which subjects would have helped you better understand the experiment.

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
