# Peer review of "Effects of a Thermal Inversion Experiment on STEM Students Learning and Application of Damped Harmonic Motion"

_sustainability, doi:10.3390/su13020919_

Round 1

Reviewer 1 Report

The manuscript is well written and its motivation has been clearly stated and developed troughout the text. 

I have the following coments:

a) The 'foreign gas' injected from below into the container is actually a 'flue gas', that is, a mixture of air and combustion products: nitrogen oxides and particles for instance. 

This point is not well worked out. One thing is to consider a pure gas composed of tiny spheres (molecules) such as in kinetic theory of gases. A quite different perspective is to visualize the motion of that 'flue gas' by tracking the combustion particles moving along with it. 

I would recommend clarifying that the experiment uses a 'flue gas' to help in visualizing its motion by tracking particles motion. Otherwise the experiment could not be carried out (unless a colored gas is injected). 

b) The value of 'r_g' seems artificial: it is too large to represent gas or particle radius in actual combustion gases. I understand it stands for the center of mass of the total amnount of particles injected into the container, represented by a single particle. This should be clarified, because I think that parameter depends on the combustion process used to generate the flue gas. 

c) Besides students in introductory physics courses, the proposed experiment is very well suited for students majoring in Earth Sciences or Environmental Engineering, because my modifying the vertical temperature gradient, different dynamic behaviors may be visualized in real time. 

Author Response

[Reviewer 1:]

[Our answer:]

We are grateful to the reviewer for the careful reading of our manuscript. In particular, the technical comments raised by the reviewer are of the utmost value to improve our manuscript and, surely, will improve the experience of the reader. We address your comments in detail in the following.

[Reviewer 1:]

I have the following comments:

  1. a) The 'foreign gas' injected from below into the container is actually a 'flue gas', that is, a mixture of air and combustion products: nitrogen oxides and particles for instance.

This point is not well worked out. One thing is to consider a pure gas composed of tiny spheres (molecules) such as in kinetic theory of gases. A quite different perspective is to visualize the motion of that 'flue gas' by tracking the combustion particles moving along with it. Correct this is the scenario.

I would recommend clarifying that the experiment uses a 'flue gas' to help in visualizing its motion by tracking particles motion. Otherwise the experiment could not be carried out (unless a colored gas is injected).

  1. b) The value of 'r_g' seems artificial: it is too large to represent gas or particle radius in actual combustion gases. I understand it stands for the center of mass of the total amnount of particles injected into the container, represented by a single particle. This should be clarified, because I think that parameter depends on the combustion process used to generate the flue gas.

[Our answer:]

Thank you for your comment. The new version of the manuscript now includes an explicit description of this as soon as possible (statements in blue in Sec. 4).

[Reviewer 1:]

  1. c) Besides students in introductory physics courses, the proposed experiment is very well suited for students majoring in Earth Sciences or Environmental Engineering, because my modifying the vertical temperature gradient, different dynamic behaviors may be visualized in real time.

[Our answer:]

Thank you so much for this suggestion. We have added these to the possible scope and applications in the conclusion (statement in blue in Sec. 8)

Again, we are grateful to the reviewer for the valuable comments and hope the new version of the manuscript addresses his/hers concerns.

Reviewer 2 Report

Dear Authors, 

the topic of your article is very interesting.

I have a few suggestions:

The introduction part is weak, please add more insight into STEM, physics teaching conceptual understanding, etc. e.g.:

  1. Freeman et al. Active learning increase student performance in science, engineering and mathematics. Proceedings of the National Academy of Sciences of the United States of America, 111(23).
  2. ME Sanders, STem, stem education, STEMmania. 2008
  3. Bybee, RW. Advancing STEM education: A 2020 vision. Technology and engineering teacher.
  4. Hockicko et al. Development of students’ conceptual thinking by means of video analysis and interactive simulations at technical universities European Journal of Engineering Education, 2015.
  5. Hake, R Interactive-engagement versus traditional methods: A six-thousand-student survey of mechanics test data for introductory physics courses. Am. J. P. 1998

Part 3 Literature review is excellently written, part Experimental methods too.

Parts 5, 6 are again weak. Please check some references about research methods for pedagogy.

Author Response

[Reviewer 2:]

Dear Authors, the topic of your article is very interesting.

[Our answer:]

Dear reviewer, thank you for your support and the careful reading of the manuscript. We are grateful for the literature suggestion. We address your suggestions in detail in the following.

[Reviewer 2:]

I have a few suggestions:

The introduction part is weak, please add more insight into STEM, physics teaching conceptual understanding, etc. e.g.:

  1. Freeman et al. Active learning increase student performance in science, engineering and mathematics. Proceedings of the National Academy of Sciences of the United States of America, 111(23).
  2. ME Sanders, STem, stem education, STEMmania. 2008
  3. Bybee, RW. Advancing STEM education: A 2020 vision. Technology and engineering teacher.
  4. Hockicko et al. Development of students’ conceptual thinking by means of video analysis and interactive simulations at technical universities European Journal of Engineering Education, 2015.
  5. Hake, R Interactive-engagement versus traditional methods: A six-thousand-student survey of mechanics test data for introductory physics courses. Am. J. P. 1998

[Our answer:]

We are grateful to the reviewer for bringing forward these references. They are now part of the bibliography in the new version of our manuscript.

[Reviewer 2:]

Part 3 Literature review is excellently written, part Experimental methods too.

[Our answer:]

Thank you so much.

[Reviewer 2:]

Parts 5, 6 are again weak. Please check some references about research methods for pedagogy.

[Our answer:]

We made some improvements of our section 5 “Qualitative Research Methods” according to your feedback. We focused our work on explaining our methodology in a better way. Most of our qualitative methods were explained further aiming to clarify possible doubts, and we added a couple of references to support our data collection and analysis process.

Although we decided to keep the section 6 “Qualitative Results” the same way to present our findings in a clear way and facilitate the reader understanding of this section, we straighten section 7 “Discussion” trying to address your feedback. This section was further analyzed and we added some comments that compare our findings with prior studies, and support most of the discussion with literature from international journals.      

You can find our changes in blue in Sec. 5 to Sec. 7.

Reviewer 3 Report

The paper is seeking to understand how students learn and apply new knowledge through open-ended activities that require mathematical modeling and experimentation focused on environmental problems. To this end, the article proposes we propose a novel home experiment for undergraduate physics courses of which results could be used to improve diverse teaching methodologies to promote both collaborative and individual work for students enrolled in STEM majors.

The structuring of the article is good. Good and relevant description of the experiment study conducted. Good presentation of the results. Results are relevant to support the improvement of the teaching methodologies. 

Author Response

[Reviewer 3:]

The paper is seeking to understand how students learn and apply new knowledge through open-ended activities that require mathematical modeling and experimentation focused on environmental problems. To this end, the article proposes we propose a novel home experiment for undergraduate physics courses of which results could be used to improve diverse teaching methodologies to promote both collaborative and individual work for students enrolled in STEM majors.

The structuring of the article is good. Good and relevant description of the experiment study conducted. Good presentation of the results. Results are relevant to support the improvement of the teaching methodologies.

[Our answer:]

We are grateful to the reviewer for the time spent reviewing our manuscript and the expressed support.

Round 2

Reviewer 2 Report

Authors improved their manuscript. I suggest accept it.